# A Binary Mixture of Emamectin Benzoate and Chlorantraniliprole Supplemented with an Adjuvant Effectively Controls *Spodoptera frugiperda*

**DOI:** 10.3390/insects13121157

**Published:** 2022-12-15

**Authors:** Junteng Zhang, Jianjun Jiang, Kan Wang, Yixi Zhang, Zewen Liu, Na Yu

**Affiliations:** 1Key Laboratory of Integrated Management of Crop Diseases and Pests (Ministry of Education), College of Plant Protection, Nanjing Agricultural University, Nanjing 210095, China; 2Guangxi Key Laboratory of Biology for Crop Diseases and Insect Pests, Plant Protection Research Institute, Guangxi Academy of Agricultural Sciences, Nanning 530007, China

**Keywords:** fall armyworm *Spodoptera frugiperda*, emamectin benzoate, chlorantraniliprole, synergism, adjuvant, field trial

## Abstract

**Simple Summary:**

The fall armyworm (FAW) *Spodoptera frugiperda* causes severe crop loss worldwide, urging effective prevention and control strategies. This study selects an effective binary mixture of emamectin benzoate and chlorantraniliprole at a mass ratio of 9:1. An oil-based adjuvant reduces the dose of the binary mixture by 80% in a field trial. This provides an effective insecticide–adjuvant mixture to control FAW.

**Abstract:**

The fall armyworm (FAW) *Spodoptera frugiperda* is a notorious pest, causing severe crop damage worldwide and prompting effective prevention and control. Over-reliance on and intensive use of insecticides are prone to leading to the rapid evolution of insecticide resistance, urging rational insecticide application. One effective way of rational insecticide application is to apply insecticides of different modes of action in combination or supplemented with adjuvants. In this study, we assessed the efficacies of two individual insecticides, emamectin benzoate (EB) and chlorantraniliprole (CT), and their mixture, supplemented with and without the oil adjuvant Jijian^®^ to control FAW in laboratory bioassays and a field trial. Both EB and CT showed high toxicities to FAW. The EB × CT mixture at a mass ratio of 9:1 yielded a remarkable synergistic effect, with the co-toxicity coefficient (CTC) being 239.38 and the median lethal concentration (LC50) being 0.177 mg/L. In leaf-spray bioassays, the addition of the adjuvant reduced the LC50 values of both the individual insecticides and the EB × CT mixture by more than 59%, significantly improving the efficacies. The field trial confirmed the synergistic effects of the adjuvant, which reduced the amount of EB × CT mixture by 80%. This study provides an effective and promising insecticide–adjuvant mixture to control *S. frugiperda*.

## 1. Introduction

The fall armyworm (FAW) *Spodoptera frugiperda* (J.E. Smith, 1797) is one of the most destructive agricultural pests worldwide. As a polyphagous migratory pest with strong reproductive capacity, FAW has caused substantial economic damage to various important crops, including maize, cotton and soybean [1]. FAW has rapidly spread in China since its first capture in Yunnan province, southwest China, in December 2018 [2]. By February 2021, FAW had been reported in 27 provinces (municipalities) and posed a threat to the corn production of about 13 million hectares in China [3]. Therefore, effective prevention and control strategies are urgently needed.

Because *Bt* transgenic maize is currently not available in China, chemical control is the major method of managing FAW in China due to its efficiency, quick effect and convenience [4,5,6,7] compared with cultural and biological controls. The Chinese Ministry of Agriculture and Rural Affair (MARA) officially recommended insecticides of different modes of action to control FAW, including spinetoram, indoxacarb, emamectin benzoate (EB) and chlorantraniliprole (CT) [8]. Because FAW has developed resistance to many types of chemical insecticides worldwide, potent insecticides should be carefully selected based on resistance status in the field [9,10,11,12]. Systemic resistance monitoring in 2019–2021 reported that FAW field populations in China showed various levels of susceptibility to the chemical insecticides recommended by MARA [13]. In general, 16 tested FAW field populations were susceptible to both CT and EB, with resistance ratios (RRs) of 0.32–2.32 for CT and 0.83–4.67 for EB. The mutation frequency was very low (0.14%–2%) in the target gene ryanodine receptor (*RyR*) of CT [13,14]. No resistance-associated mutations in the target gene, the glutamate-gated chloride channel *GluCl* of EB, were detected in a total of 2806 individuals collected in 2019–2021 [13]. However, some FAW populations in 2021 exhibited a slightly higher RR to CT than that of populations in 2019, indicative of the risk of resistance development. Two populations exhibited low levels of resistance to EB, which might have been caused by higher spraying frequencies and intensive use of EB in the regions [13]. The risk of resistance to EB and CT in some field populations has been attributed to increased detoxification and certain ABC transporters [15,16]. Therefore, rational application of emamectin benzoate and chlorantraniliprole should be considered to reduce the risk of resistance development [17].

A binary mixture of insecticides improves the pest controlling performance of both individual insecticides, requiring a lower amount of insecticides and leading to reduced risk of insecticide resistance development. A great number of insecticide mixtures have been applied to control pests in the field. A mixture of spinosad and indoxacarb at a ratio of 1:9 showed a significant synergistic effect against FAW with a co-toxicity coefficient (CTC) of 147 in a diet-incorporated bioassay [18]. A mixture of EB and tetrachloropamide at a ratio of 7:3 was the most effective, with a CTC of 162 in a leaf-dipping bioassay [19]. Emamectin benzoate 5% EC mixed with acephate 75% SP had strong toxicity to the 2nd instar FAW larvae, with a corrected mortality rate of over 90% [20].

Therefore, it is necessary to formulate reasonable pesticide combinations and to search for an effective adjuvant to guide the scientific use of pesticides and to extend the service life of pesticides in the field.

## 2. Materials and Methods

### 2.1. Insect Rearing and Insecticides

*S. frugiperda* larvae were reared individually with an artificial diet at 27 ± 1 °C, 65 ± 5% relative humidity and a photoperiod of 16 h:8 h light: dark cycle in a climate chamber [13]. The larvae were individually housed in a polystyrene cup (4.5 cm high, 3.5 cm diameter) with a lid in order to avoid cannibalism. Pupae were sexed and kept in cages for eclosion. Moths were fed with a 10% honey solution.

The insecticides used in the bioassay included emamectin benzoate (90%), chlorantraniliprole (97%), beta-cypermethrin (95%), diflubenzuron (97%) and pleocidin (90%), all purchased from Sunlida Biotechnology Co., Ltd. (Nanjing, China), as well as 5% emamectin benzoate WDG (5% emamectin benzoate in the form of water-dispersible granules, Sichuan Jinzhuang Technology Co., Ltd., Chengdu, China) and 200 g/L chlorantraniliprole SC (Coragen^®^ 200 g/L chlorantraniliprole in the form of a suspension concentrate, FMC, Suzhou, China). The adjuvant Jijian^®^ was a commercial product provided by Chengdu Jijian Biotechnology Co., Ltd. (Chengdu, China).

### 2.2. Diet-Incorporated Bioassay

The lethal-dose effects of individual emamectin benzoate (EB), chlorantraniliprole (CT) and their serial mixture (EB × CT mixtures) were determined with diet-incorporated insecticide bioassays following the procedures described in [21] with modifications. The insecticides were dissolved in acetone as stock solutions. Each stock solution was diluted to five serial concentrations with water. EB and CT were mixed at mass ratios (EB:CT) of 1:9, 3:7, 5:5, 7:3 and 9:1. An amount of 15 mL of the diluted insecticide or mixture solution was added to 85 mL of artificial diet and was agitated for 75–90 s in a 1 L bowl with a hand mixer. Approximately 2 g of the insecticide-incorporated diet was placed in each well of a 24-well plate, and 4 wells were prepared for each concentration. A total of 10 3rd-instar larvae (L3, weight 6.2–7.5 mg/larva) were placed in each well, and a total of 40 larvae were used for each concentration. Diets containing the same volume of acetone: water were fed to the control larvae. Insect mortalities were recorded after 72 h. The larvae were considered dead if they were unable to move when prodded with a soft brush.

### 2.3. Leaf-Spray Bioassay

The lethal-dose effects of individual EB and CT and the EB × CT mixture (9:1) supplemented with or without Jijian^®^ (0.1%) were determined using the leaf-spray method. EB, CT and EB × CT mixture solutions of five serial concentrations were prepared, and 30 mL of each concentration was used in the maize leaf treatment. Maize leaves from plants in the jointing stage were cut into 5 cm-long pieces and were put on a home-made tray. The tray supported the leaves in a position 60° to the horizon, thus forming a natural maize leaf position (Appendix A). An amount of 30 mL of the insecticide solution was sprayed onto 16 leaves in a spray tower (3WPSH-500D, Nanjing Institute of Agricultural Mechanization, Nanjing, China) with the chassis spindle speed at 6 rpm in the top spray mode and with the pressure regulating valve at 3 kg/cm^2^. After spraying, maize leaves were allowed to dry on the bench. Four leaf pieces were put in one Petri dish (11 cm diameter, 1.5 cm high), and four Petri dishes were prepared for each concentration. A total of 10 L3 FAW larvae (body weight 6.2–7.5 mg/larva) were placed into one Petri dish, and a total of 40 larvae were used for each concentration. Insect mortalities were recorded after 72 h. Larvae were considered dead if they were unable to move when prodded with a soft brush.

The synergistic effect of adjuvant Jijian^®^ on insecticides was also evaluated with diflubenzuron (10 mg/L), pleocidin (10 mg/L) and beta-cypermethrin (100 mg/L) via the leaf-spray bioassay following the same setups. Insect mortalities were recorded after 48 h.

### 2.4. Field Trial

The efficacies of EB, CT and the EB × CT mixture (9:1) supplemented with or without Jijian^®^ (0.1%) were evaluated in a field trial. The field trial was carried out in the experimental field of Guangxi Academy of Agricultural Sciences, Guangxi province, Southern China. The whole maize field, including experimental plots and cushion area, was fertilized and managed in a standard maize culture routine. Each experimental plot covered an area of 30 m^2^. The plots were randomly designated for the treatment or control groups. The maize variety Chuanhe 968 was sown on 28 September 2021. The maize density was 50,000–60,000 plants/hectare, with a row spacing of 0.6 m and a plant spacing of 0.3 m within a row. The experiment was conducted when maize plants were at the jointing stage and when the FAW infestation was serious.

The treatments included 5% emamectin benzoate WDG (5% EB, 15 g/ha.), 200 g/L chlorantraniliprole SC (200 g/L CT, 150 mL/ha.), EB × CT I (routine dose of EB × CT mixture at 9:1, 5% EB 13.5 g/ha. and 200 g/L CT 0.375 mL/ha.), EB × CT II (reduced dose of EB × CT mixture at 9:1, 5% EB 2.7 g/ha. and 200 g/L CT 0.075 mL/ha.) and EB × CT II × adjuvant (reduced dose of EB × CT mixture at 9:1 supplemented with adjuvant, 5% EB 2.7 g/ha., 200 g/L CT 0.075 mL/ha., and 450 mL/ha. Adjuvant Jijian^®^). Water (450 L/ha.) was sprayed for the control.

At least one FAW larva per maize plant was observed in the sampling in each experimental plot on 6 November 2021. The insecticides or water were sprayed at 0.95 L/min using a 3WBJ-16 electric knapsack sprayer (Xiping Chilong Plant Protection Machinery Co., Ltd., Zhumadian, China) with a moving speed of 11.25 m/min on the evening of 7 November 2021. The weather conditions on 7 November were suitable for insecticide spraying and insect examination, with a temperature of 28.6 °C, a relative humidity of 71.3% and a wind speed of 0.3 m/s. During the trial, it did not rain, and the temperature range was 16–30 °C. A two-meter-wide buffer zone was set up on each side of the plot to avoid spray drift.

In each plot, 17–26 maize plants were marked and checked for the numbers of living FAW on day 1, day 3 and day 7 post spraying. The inhibition of emergence (EI) was calculated following Henderson–Tilton’s formula, as follows: $\mathrm{EI}=\left(1-\frac{T_{a}\times C_{b}}{T_{b}\times C_{a}}\right)\times 100\%$, where *T_a_* and *T_b_* are the pre- and post-treatment pest densities (the number of individuals collected per sampling effort) in the treated fields, respectively, and *C_a_* and *C_b_* are the pest densities in control fields at the same times, respectively [22].

### 2.5. Statistical Analysis

The data of the laboratory bioassays were corrected following Sun et al. [23] and were then subjected to probit analysis in Polo-Plus [24]. The LC_50_ values of individual insecticides, the insecticide mixtures of various mass ratios and the insecticide supplemented with the adjuvant were calculated. Statistical significance was determined via a *t* test or a one-way ANOVA followed by Duncan’s new multiple range test at *p* < 0.05 using GraphPad Prism version 9.0.0 for Windows (GraphPad Software, San Diego, CA, USA, www.graphpad.com (accessed on 30 September 2022)). The co-toxicity coefficients (CTCs) of the insecticide mixtures were calculated and used to evaluate the synergistic effect following a previously reported method [25]. CTC values greater than 120 represent a synergistic effect, a CTC value less than 80 shows an antagonistic effect and a CTC value of 80–120 suggests an additive effect.

## 3. Results

### 3.1. Synergistic Effect of the EB × CT Mixture

The LC_50_ values of EB and CT at 72 h were 0.383 mg/L and 9.703 mg/L, respectively, in the diet-incorporated bioassay (Table 1). When mixed at a ratio of 9:1, EB × CT yielded the highest toxicity against FAW, with an LC_50_ of 0.177 mg/L, and the highest synergistic effect, with a CTC of 239.38. EB × CT (3:7) also exhibited impressive synergistic effects, with a CTC of 128.74. Additive effects were observed for EB × CT at ratios of 5:5 and 7:3, and antagonist effects were seen in EB × CT at a ratio of 1:9 (Table 1).

### 3.2. Synergistic Effect of Adjuvant JIJIAN^®^ on Insecticides and the EB × CT Mixture

The lethal effects of EB, CT and the EB × CT mixture supplemented with or without the adjuvant Jijian^®^ in the leaf-spray bioassay were calculated, as shown in Table 2. The LC_50_ values of individual EB and CT were 0.842 mg/L and 35.173 mg/L, respectively. The LC_50_ values of the EB × CT mixture (9:1) was 0.482 mg/L. The addition of the adjuvant Jijian^®^ reduced the LC_50_ of EB, CT and the EB × CT mixture from 0.842 mg/L to 0.310 mg/L, from 35.173 mg/L to 14.200 mg/L and from 0.482 mg/L to 0.197 mg/L, respectively, showing strong synergistic effects. The LC_90_ value of EB × CT × Jijian^®^ of 0.930 mg/L accounted for half of the LC_90_ of the EB × CT mixture alone (2.049 mg/L). In addition, the addition of Jijian^®^ significantly increased the lethality of insecticides of different modes of action compared with the insecticide alone (Appendix A).

### 3.3. Promising Efficacy of EB × CT × Jijian^®^ in the Field Trial

The results of our field trial are summarized in Figure 1 and Appendix A. The 5% EB WDG alone yielded the highest and most rapid control of FAW, with an inhibition of emergence (EI) of 80–91% within 7 days after insecticide spraying. An amount of 200 g/L CT SC was less effective than other treatments on day 1, but the FAW controlling effectiveness rose steeply on day 3 and reached an equivalent effectiveness to the other treatments on day 7. EB × CT I showed a moderate and steadily increasing effect over the period and achieved a similar controlling effect to those of individual EB and CT. With only 20% of the insecticide amount of EB × CT I, EB × CT II was less effective than individual EB, CT and EB × CT I, with an EI range of 70–89%. However, the EB × CT II × adjuvant controlled FAW as quickly and effectively as individual EB and CT, with its EI increasing from 71% on day 1 to 84% on day 7.

## 4. Discussion

Chemical control is currently a primary and effective measure to control FAW in China, but the risk of insecticide resistance development should be taken seriously when long-term control strategies are considered. Because of the invasion of FAW in China, systemic resistance monitoring on commonly used insecticides has been conducted throughout the nation [26,27]. In general, most tested FAW field populations are susceptible or exhibit low resistance to both CT and EB [13,14,16]. Therefore, measures should be taken to prevent insecticide resistance. One reliable way of preventing insecticide resistance is to apply insecticides of different modes of action in a mixture or rotationally. A great number of insecticide mixtures have been evaluated for their efficacy against FAW in the laboratory or in the field. Hu et al. [28] reported that a mixture of EB and CT at a ratio of 3:7 showed the highest synergism, with a CTC of 173 in a larva-dipping bioassay using the 2nd FAW larvae. In the present study, an EB × CT mixture at a ratio of 9:1 displayed an excellent controlling effect on FAW both in the laboratory and in the field. We attribute the discrepancy to the different methods and tested larvae used in the bioassays. Nevertheless, the binary mixture of EB and CT has been an effective insecticide combination for controlling FAW because the field control efficacies in both the study of Hu et al. [28] and the present study are over 88%.

Rational use of adjuvants can significantly reduce the use of pesticides. The oil-based adjuvant Jijian^®^ has been used in rice, cotton and tea fields in China. The addition of Jijian^®^ reduced the amount of pymetrozine × etrofolan by 30–40% in controlling planthoppers in rice fields [29], reduced the amount of imidacloprid by 50% in controlling aphids [30], and reduced the amount of CT in controlling *Ostrinia furnacalis* in maize fileds [31]. In the present study, Jijian^®^ reduced the LC_50_ values of EB, CT and the EB × CT mixture by 59% in the leaf-spray bioassay in the laboratory, and it further reduced the amount of EB × CT mixture by 80%, with an equivalent efficacy in controlling FAW as that of the EB × CT mixture in the maize fields (Table 2, Figure 1). In addition, Jijian^®^ was effective in reducing the amount of insecticides of distinct groups (Table 2, Appendix A). The exact mechanism of the remarkable effect of Jijian^®^ on reducing the amount of the EB × CT mixture requires further study.

The field trial in the present study demonstrates that the EB × CT mixture supplemented with adjuvant Jijian^®^ provides an effective control method against FAW that could be further implemented in the field. The binary mixture and the adjuvant, as well as other kinds of adjuvants, could further be exploited and expanded to controlling a vast range of pest insects.

## Figures and Tables

**Figure 1 insects-13-01157-f001:**
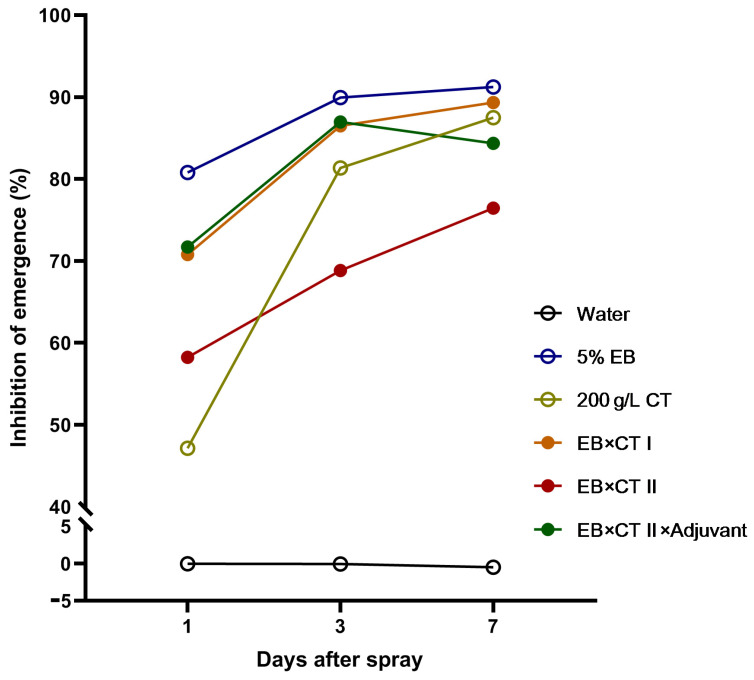
Efficacies of the emamectin benzoate × chlorantraniliprole mixture with and without adjuvant Jijian^®^ against *S. frugiperda* in a field trial. Efficacies were assessed by the mean of the inhibition of emergence (EI, %) of insect populations sampled in four trial plots. EB, emamectin benzoate; CT, chlorantraniliprole. Water, 450 L/ha.; 5% EB, 5% emamectin benzoate WDG, 15 g/ha.; 200 g/L CT, 200 g/L chlorantraniliprole SC, 150 mL/ha.; EB × CT I, routine dose of EB × CT mixture at 9:1, 5% EB, 13.5 g/ha., and 200 g/L CT, 0.375 mL/ha.; EB × CT II, reduced dose of EB × CT mixture at 9:1, 5% EB, 2.7 g/ha., and 200 g/L CT, 0.075 mL/ha.; EB × CT II × adjuvant, reduced dose of EB × CT mixture at 9:1 supplemented with adjuvant, 5% EB 2.7 g/ha., 200 g/L CT 0.075 mL/ha. and Adjuvant Jijian^®^ 450 mL/ha.

**Table 1 insects-13-01157-t001:** Synergistic effect of emamectin benzoate and chlorantraniliprole in diet-incorporated bioassays.

Pesticides ^a^	Mass Ratio (EB:CT)	LC_50_ (mg/L)	95% Confidential Limit	Slope ± SE	Chi-Square	ATIM ^b^	TTIM ^b^	CTC ^b^
EB	/	0.383	0.280~0.500	1.853 ± 0.273	1.950	/	/	/
CT	/	9.703	5.906~15.012	1.169 ± 0.243	1.118	/	/	/
EB × CT	1:9	5.390	3.305~10.114	1.259 ± 0.296	0.341	7.11	13.55	52.43
	3:7	0.908	0.631~1.348	1.662 ± 0.314	2.659	42.18	32.76	128.74
	5:5	0.620	0.497~0.773	2.381 ± 0.305	2.603	61.77	51.97	118.86
	7:3	0.659	0.498~0.857	2.146 ± 0.322	0.587	58.12	70.18	81.65
	9:1	0.177	0.129~0.231	2.354 ± 0.410	0.191	216.38	90.39	239.38

^a^ EB, emamectin benzoate. CT, chlorantraniliprole. ^b^ ATIM, actual virulence index. TTIM, theoretical virulence index. CTC, co-toxicity coefficient. /, not applicable.

**Table 2 insects-13-01157-t002:** Synergistic effect of Jijian^®^ adjuvant on the toxicity of emamectin benzoate, chlorantraniliprole and EB × CT in leaf-spray bioassays.

Insecticides or Mixture ^a^	LC_50_ (mg/L)	95% Confidential Limit	Slope ± SE	Chi-Square	ATIM ^b^	TTIM ^b^	CTC ^b^
EB	0.842	0.678~1.080	2.259 ± 0.303	2.213	/	/	/
EB × adjuvant	0.310	0.195~0.544	1.832 ± 0.258	3.569	/	/	/
CT	35.173	26.580~45.94	1.711 ± 0.250	0.754	/	/	/
CT × adjuvant	14.200	11.364~17.765	2.413 ± 0.368	1.193	/	/	/
EB × CT (9:1)	0.482	0.380~0.609	2.039 ± 0.269	2.213	174.69	90.24	193.58
EB × CT (9:1) × adjuvant	0.197	0.151~0.252	1.903 ± 0.262	3.569	157.36	90.22	174.42

^a^ EB, emamectin benzoate; CT, chlorantraniliprole; adjuvant, Jijian^®^. ^b^ ATIM, actual virulence index; TTIM, theoretical virulence index; CTC, co-toxicity coefficient; /, not applicable.

## Data Availability

The data are contained within this article or in the supplementary materials.

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
