# Peer review of "A Binary Mixture of Emamectin Benzoate and Chlorantraniliprole Supplemented with an Adjuvant Effectively Controls Spodoptera frugiperda"

_insects, 2022, doi:10.3390/insects13121157_

Round 1
Reviewer 1 Report
The manuscript entitled “A binary mixture of emamectin benzoate and chlorantraniliprole supplemented with an adjuvant controlled Spodoptera frugiperda effectively” reports an insecticide mixture and an adjuvant, both effective on controlling the notorious insect pest S. frugiperda. The findings here were straightforward and are particularly useful in the rational application of insecticides to control S. frugiperda.
I would suggest the authors to revise the manuscript to improve its readability. Here are some considerations.
1. Spodoptera frugiperda should be italic throughout the manuscript.
2. Based on the context, I would suggest to combine lines 63-72 into one single paragraph.
3. Line 14, mamectin benzoate should be emamectin benzoate
4. Line 95, followed should be following.
5. Reference list should be carefully formatted.
Reviewer 2 Report
The manuscript “A binary mixture of emamectin benzoate and chlorantraniliprole supplemented with an adjuvant controlled Spodoptera frugiperda effectively” found an effective binary mixture of emamectin benzoate and chlorantraniliprole at mass ratio of 9:1 and the effectiveness of the oil adjuvant Jijian has also been verified, provided an effective control method against FAW that could be further implemented in field. However, there are some problems to be addressed in the current version before it can be accepted for publication.
1. Line 101-102. Because of cannibalism, I think 10 fall armyworms in one well is too crowded.
2. Line 117. The size of the Petri dish should be clear.
3. There are so many adjuvants in market, only Jijian has been determined, why?
4. Line 215. “S. frugiperda” should be italic, please double-check all the similar typos.
5. Line 213. Figures and Tables should be self-explained, i.e., some explanations should be available in Notes of figures for abbreviations or other necessary terms, such as EB x CT I, EB x CT II, EB x CT II x Adjuvant etc., so that readers can understand most of the meanings without reading the text.
6. Line 14. “mamectin benzoate” should be “emamectin benzoate”.
7. In your discussion, Jijian® reduced the amount of EB×CT mixture by 80% with the equivalent efficacy in controlling FAW as the EB×CT mixture in the maize field, but in Fig 1, 5% EB WDG alone yielded the rapidest and highest controlling of FAW with the inhibition of emergence (EI) 80%-91% within 7 days after insecticide spray, how can you explain this conflict?
Reviewer 3 Report
On the attached file, I have some minor observations in the introduction, and materials and methods; also I have some suggestions in Discussion.
